# Low Molecular Weight Hyaluronic Acid (500–730 Kda) Injections in Tendinopathies—A Narrative Review

**DOI:** 10.3390/jfmk7010003

**Published:** 2021-12-29

**Authors:** Antonio Frizziero, Filippo Vittadini, Davide Bigliardi, Cosimo Costantino

**Affiliations:** 1Department of Medicine and Surgery, University of Parma, 43126 Parma, Italy; davide.bigliardi@unipr.it (D.B.); cosimo.costantino@unipr.it (C.C.); 2Department of Physical and Rehabilitation Medicine, Casa di Cura Policlinico S. Marco, 30100 Venice, Italy; filippo.vittadini@gmail.com

**Keywords:** tendinopathy, hyaluronic acid, injection

## Abstract

Tendinopathies are common causes of pain and disability in general population and athletes. Conservative treatment is largely preferred, and eccentric exercise or other modalities of therapeutic exercises are recommended. However, this approach requests several weeks of consecutive treatment and could be discouraging. In the last years, injections of different formulations were evaluated to accelerate functional recovery in combination with usual therapy. Hyaluronic acid (HA) preparations were proposed, in particular LMW-HA (500–730 kDa) for its unique molecular characteristics in favored extracellular matrix homeostasis and tenocyte viability. The purpose of our review is to evaluate the state-of-the-art about the role of 500–730 kDa in tendinopathies considering both preclinical and clinical findings and encourage further research on this emerging topic.

## 1. Introduction

Tendinopathy is multifactorial pathology, that could lead to chronic pain and functional impairment [1]. The rotator cuff [2], the long head of the biceps [3], the wrist extensors and flexors [4], the adductors [5], the patellar tendon, the Achilles tendon [6] and the posterior tibial tendon [7] are the most affected tendons.

Tendinopathies are largely triggered by overload [8], but a great number (about one third) of patients do not practice regular physical activity [9].

Tendinopathies can occur in acute and chronic state and can be supported by different factors both intrinsic (age [10], body structure [11], nutrition [12], metabolic diseases [13,14], genetics [15]) and extrinsic (excessive and improper fatigue [8,16], detraining [17], and external damage [18,19,20]) factors. All these factors seem to be related to the appearance of overload tendinopathy although a cause–effect relationship has not yet been demonstrated.

Tendinopathies are characterized by disorganization of collagen fibers with dysregulation in extracellular matrix homeostasis, alteration in proteoglycan content, enhanced tenocyte apoptosis, and increase in the microvasculature and sensory nerve innervations [19,21]. Additionally, paratenon shows mucoid degeneration, fibrosis, and vascular proliferation with mild inflammatory infiltrate [22].

There are currently no treatments of choice for tendinopathies as no strong evidence has yet been found that one treatment is better than another [1,23,24]. Until a few years ago, eccentric exercise [25,26,27] and extracorporeal shock wave therapy (ESWT) [28] showed significant effects on both pain and functional recovery.

Recently, thanks also to the ultrasound high-definition evaluation and guide [29], injection therapy has also carved out an important role in the treatment of tendinopathies through different proposals with different levels of success, as hyaluronic acid [30], PRP [31], high volume injections [32], prolotherapy [33,34], autologous conditioned serum [35], sclerotherapy [34], percutaneous electrolysis [36], etc.

Hyaluronic acid (HA) is a polymer of disaccharides, which are composed of d-glucuronic acid and *N*-acetyl-d-glucosamine, linked via alternating β-(1→4) and β-(1→3) glycosidic bonds [37].

In particular, HA allows a smooth tendon gliding and provides nutrition to tendon itself, and it is an important component of tendon structure, being largely present in extracellular space.

HA can be classified by origin (rooster combs or bacterial fermentation), molecular organization (linear, cross-linked, or hybrid), and molecular weight [38]:Low molecular weight: <1000 KDa;Medium molecular weight: 1000 KDa < MW < 2000 KDa;High molecular weight: >2000 KDa.

Most of the physiological effects of exogenous hyaluronic acid and its viscosity depend primarily on the molecular weight of the molecule, consequently influencing its possible applications [30].

Low molecular weight hyaluronic acid (500–730 KDa) formulations were the first proposed in clinical practice for knee osteoarthritis since 1990, considering “viscosupplementation” as purpose with high safety profile [39,40,41]. However, several preclinical and clinical studies increased the knowledge about the numerous possible effects of the interaction with articular tissues as cartilage subchondral bone, synovial membrane, and most recently on tendons.

In recent years, the role of low molecular weight HA (LMW-HA) is emerging among the possible treatments of tendon pathology, secondary to the growing scientific evidence in the preclinical field and to the greater sophistication of ultrasound and techniques to perform therapeutic interventions safely [42].

The purpose of the present study is to perform a descriptive review to highlight the state of the art on the efficacy of hyaluronic acid at low-molecular weight injections in the treatment of tendinopathies.

## 2. Materials and Methods

The present systematic review was conducted on common electronic databases including PubMed (MEDLINE), Pedro, and Cochrane Library. Each database was searched according to its specific syntax rules. Moreover, manual search of published and unpublished studies (conference abstracts, textbooks, ‘grey’ literature) was also conducted and reference lists of retrieved articles were screened. Search results were limited to studies written in English and published until August 2021. Studies were selected using the following keywords alone or in combination: “hyaluronic acid”, “low molecular weight hyaluronic acid 500–730 KDa”, “tendinopathy”, “tendon” and “injection”. The search results and the relative selection process are shown in the following flowchart (Figure 1).

### Study Selection

The selection criteria were: (i) language of publications (English); (ii) study design (preclinical studies, RCT or prospective studies); (iii) sample, including adult patients with a diagnosis of Achilles, patellar or supraspinatus tendinopathy, lateral epicondylitis, or trigger digit; (iv) outcome measures, including pain and functional improvement, patient satisfaction, or return to sport; (v) treatment, consisting in hyaluronic acid injection, alone or associated with other therapies (i.e., exercise). Using the above-mentioned criteria, relevant studies were selected after reading the title and/or abstract, while publications that did not meet the criteria were excluded. Full text from articles selected for inclusion in the review was read, and the bibliographies were hand searched for further relevant publications. As a result, some studies were withdrawn after reading the full text, whereas others were retrieved from the bibliography of the selected articles.

## 3. Results

The initial search yielded 76 articles, of which 34 were identified as relevant after screening of titles/abstract. Twenty records were excluded because they were not experimental studies or pre-clinical studies. Two papers were excluded after reading full article for methodological issues.

Finally, 12 papers were retrieved from literature search. Among them, 8 clinical studies were selected. Only one compared LMW-HA against placebo; 4 were RTC; and 2 published observational study (Table 1 and Table 2).

### 3.1. In Vitro Studies

Osti et al. evaluated the effects of four different hyaluronic acid formulations on viability, metabolic activity, apoptosis, and collagen type I and collagen type III expression on human rotator cuff tendon tears derived cells. The authors found that all hyaluronic acid preparation were efficient in amelioration of cell-metabolic activity, decrease rate of the apoptosis of tendon derived cells and increase Collagen I production in dose dependent manner not related to the molecular weight [43].

Gallorini et al. investigated the antioxidant effects of the same four HA formulations and concluded that all the molecules have protective effects against the oxidative stress-related cytotoxicity. However, protective effects appear to be particularly efficient in tendon cells treated with LMW-HA (500–730 KDa) [44].

### 3.2. Pre-Clinical Studies

Salamanna et al. investigated the effects of repeated peri-patellar injections of LMW-HA (500–730 KDa) in sudden-detraining rat model tendinopathy, founding that HA allow the maintenance of tenocyte anabolic activity and proliferation in comparison to saline or no intervention [45].

Furthermore, in a second study Frizziero et al. demonstrated that repeated peri-patellar injections of LMW-HA may maintain the structural and functional properties of patellar tendon and enthesis in sudden detrained rats [46].

### 3.3. Clinical Studies

Meloni et al. found that five US-guided LMW-HA injections were effective compared to placebo in improving pain and function in 56 patients affected by supraspinatus tendinosis until 9 months [47].

Blaine et al. found that LMW-HA (500 to 730 kDa) injection is effective in the short term and well tolerated for the treatment of osteoarthritis and persistent shoulder pain (including rotator cuff tear) that is refractory to other standard non operative interventions [48].

Frizziero et al. observed that LMW-HA (500 to 730 kDa) and low-energy ESWT were both effective and safe in patients suffering from non-calcific rotator cuff tendinopathy until a 3-month follow-up. However, LMW-HA provide prompt clinical improvement compared to ESWT [28].

Fogli et al. have shown that three US-guided LMW-HA paratendinous injections induce significant progressive improvement in pain and function and US parameters (tendon thickness and neovascularization) in patients with Achilles or patellar tendinopathies [49].

Similar good results in terms of pain reduction and improvement of function in Achilles and patellar tendinopathy have also been observed by Frizziero et al. following US-guided paratendinous injection treatment [50].

Kanchanathepsak et al. compared the efficacy of US-guided injection of LMW-HA (500–730 KDa) and corticosteroids (triamcinolone acetate) in the treatment of trigger finger. The authors noted that both LMW-HA and corticosteroids provide a comparable therapeutic effect in the medium term. However, it is necessary to consider that corticosteroids have a cytotoxic effect on tendons [51].

Recently, Mohebbi et al. evaluated the clinical effects of HMW-HA (>2000 kDa) versus LMW-HA (500–730 KDa) single US-guided periarticular injection in the management of rotator cuff tendinopathy with persistent shoulder pain for more than 6 weeks. The pain, induration, and inflammation at the site of injection are less prominent for LMW-HA compared with HMW-HA. Even both preparations were effective in improving pain, joint function, and quality of life that last at least for 3 months, LMW-HA is less expensive and resulted more tolerable to the patient. LMW-HA seems also more effective than physiotherapy in alleviating pain and improving shoulder ROM and quality of life. Furthermore, patient compliance was higher with LMW-HA injection in relation to lower perceived pain [52].

Rezasoltani et al. compared the effects of LMW-HA injection with physiotherapy in patients with supraspinatus tendinopathy and concluded that injection therapy showed greater efficacy at a 3-month follow-up. In particular, hyaluronic acid is more successful in relieving pain [53].

## 4. Discussion

In past years, intra-articular injections of hyaluronic acid were a well-known and widely used option in the conservative treatment of osteoarthritis with positive results. On the other hand, the role of HA in tendinopathies was underestimated. Although further studies are needed to demonstrate the clinical efficacy and safety of hyaluronic acid in the management of tendinopathies, several studies demonstrated positive effects on tendinopathies with low molecular weight hyaluronic acid injections.

The more frequent performance of ultrasound evaluation and the possibility of positioning the therapeutic substance with extreme precision in the affected site have allowed for better results in the clinical field.

The results observed in in-vitro studies have showed protective effects against the oxidative stress-related cytotoxicity in the tendon derived cells treated with the three lower molecular weight HAs [43,44].

Osti et al. identified a protective effect of the tendon by hyaluronic acid, related to the very low expression for the production of collagen type III [43].

The encouraging effects of hyaluronic acid in conditions of tendon suffering have also been confirmed in pre-clinical studies [45,46]. Salamanna et al. [45] and Frizziero et al. [46] investigated the effects of repeated peri-patellar injections of low molecular weight HA (500–730 KDa) on tenocytes morphology, viability, proliferation, and metabolic activity and both the authors concluded that HA injections at the end of the I, II, III, and IV weeks in detrained tendinopathies rat model seems to help tenocytes to maintain their synthetic and metabolic activities, the structural and functional properties of the patellar tendon in ensuring safety.

Following the results just described where LMW-HA has shown the potential to determine amelioration of tendon behavior via change in tenocyte viability and proliferation and collagen composition were promoted clinical studies on human different tendon’s site (rotator cuff tendons, patellar tendon, Achilles tendon, etc.) with encouraging conclusions [47,48,49,50,51,52,53].

In fact, the results obtained by Meloni et al. [47] in rotator cuff pathology shows better results with hyaluronic acid than placebo, while Frizziero et al. [28] demonstrated faster efficacy of injections with low molecular weight hyaluronic acid compared to shock waves.

Thus, it could be speculated that LMW-HA should contribute to the restoration of the essential viscoelastic properties of tendon, leading to function improvement without side effects.

Mohebbi et al. [52] observed in a recent comparative study between ultrasound-guided injections with low molecular weight (500–730 KDa) and high molecular weight (>2000 KDa) that, with the same effectiveness, low molecular weight was more tolerated by patients.

Rezasoltani et al. [53] and Blaine et al. [48] confirmed the efficacy of low molecular weight hyaluronic acid (500–730 KDa) injection in pain relief and function amelioration.

Similar positive results on pain and functional recovery have been observed in the clinical experiences of Frizziero et al. [50] and Fogli et al. [49] in the Achilles tendon, patellar tendon, and epicondylitis.

Pathological conditions such as tenosynovitis also benefited from injection treatment with low molecular weight hyaluronic acid (500–730 KDa) in comparison to triamcinolone acetonide. Despite the corticosteroid produced more rapid remission of the symptoms, hyaluronic acid was also shown to be safer and produced a longer lasting effect [51].

All the studies have found that clinical effectiveness should be reached already at 1 month after the LMW-HA injections; this finding should be crucial in athletes that need safe and rapid return to sport participation.

Furthermore, it appears that LMW-HA injection was highly tolerated by patients, with only rare local, self-limiting side-effects. In specific, LMW-HA presented better tolerability and minor costs in comparison to HMW-HA formulations [52].

## 5. Conclusions

Tendinopathies are a group of pathologies that afflict the tendons and nowadays there are many current treatment options, but none has shown particular superiority. Among these therapeutic proposals and following favorable results observed with in vitro, pre-clinical, and clinical studies, the use of low molecular weight hyaluronic acid by injection should be encouraged compared to high-weight hyaluronic acid. The rapid action and the high safety profile make this treatment particularly suitable for athletes. To achieve extreme accuracy of positioning of hyaluronic acid, the use of ultrasound guidance is strongly recommended. Even the lower cost compared to hyaluronic acids with higher molecular weight can represent an additional incentive to its use.

## 6. Limitations of the Review

Despite the limitations of our study represented by the low number of papers in the literature and the small number of patients treated, the results obtained encourage the use of low molecular weight hyaluronic acid in tendinopathies. For these reasons, further studies are needed to produce more evidence of efficacy of low molecular weight hyaluronic acid in tendinopathies.

## Figures and Tables

**Figure 1 jfmk-07-00003-f001:**
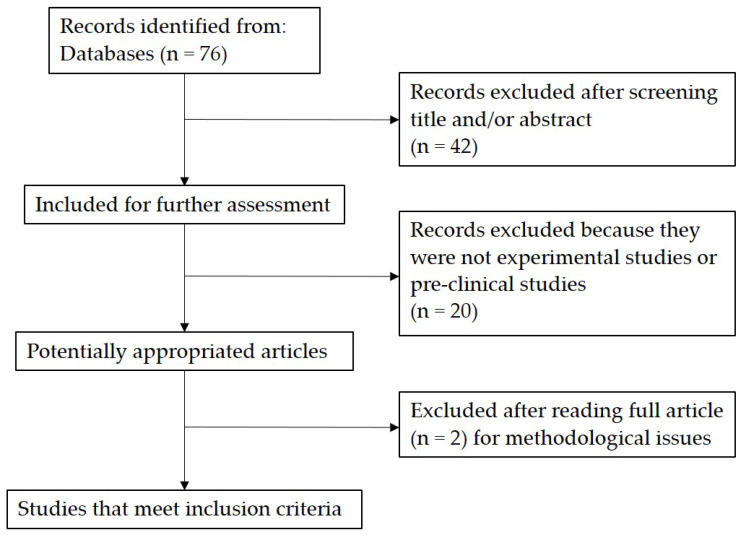
Summary of studies identification and selection.

**Table 1 jfmk-07-00003-t001:** Four papers were retrieved from literature search. Among them, 2 in vitro studies, 2 pre-clinical studies were selected.

**In Vitro Studies**				
**Article**	**Sample Description**	**Experimental Group**	**Control Group**	**Outcome**
Osti et al., 2015	Human rotator cuff tears tendon-derived cells	250 μg/mL GROUP-MW 500–730 KDa, 500 μg/mL GROUP-MW 500–730 KDa, 1000 μg/mL GROUP-MW 500–730 KDa, 250 μg/mL GROUP-MW 1000 KDa, 500 μg/mL GROUP-MW 1000 KDa, 1000 μg/mL GROUP-MW 1000 KDa, 1000 μg/mL GROUP-MW 1600 KDa, 1000 μg/mL GROUP-MW 2200 KDa,	Untreated cells	Tenocyte viability and proliferation, Apoptosis induction, Immunofluorescence staining
Gallorini et al., 2019.	Human supraspinatus tendon-derived cells	H_2_O_2_ GROUP: cultured cells were exposed to 2 mM H_2_O_2_ 1000 KDa MW GROUP 2200 KDa MW GROUP 500–730 KDa MW GROUP 1600 KDa MW GROUP H_2_O_2_^+^ HA GROUPS: cells were treated with different hyaluronic acids in the presence of 2 mM H_2_O_2_	Untreated cells	Cytotoxicity-lactate dehydrogenase (LDH) release assay, immunofluorescence staining of CD4, measurement of mitochondrial membrane potential (TMRE assay) by flow cytometry, cell lysis and protein extraction, immunoblotting, catalase activity
**Pre-Clinical Studies**				
**Article**	**Sample Description**	**Experimental Group**	**Control Group**	**Outcome**
Salamanna et al., 2014	24 male Sprague-Dawley rats, aged 8 weeks, 280 ± 40 g body weight, Patellar tendon 18 rats run on a treadmill 1 h a day, three times a week for 10 weeks. The other six rats underwent no training	TRAINED GROUP (6 rats)–euthanized at the end of the 10-week training. Detrained group (12 rats)-euthanized after 10-week training + 4 weeks without exercise DETRAINED-HA GROUP (6 of the 12 detrained rats)-at the end of the 1, 2, 3, and 4 weeks without exercise a peri-patellar infiltration of 300 μL of 20 mg/2 mL HA (500–730 KDa) was injected in the right patellar tendon DETRAINED-NaCl GROUP (remaining 6 of the 12 detrained rats)-at the end of the 1, 2, 3, and 4 weeks without exercise, at the level of the right patellar tendon, received a peri-patellar infiltration of 300 μL of physiological saline solution (Fresenius Kabi)	UNTRAINED GROUP (6 rats)-euthanized without training	Tenocyte morphology and morphometric analysis (transmission electron microscopy), tenocyte viability and proliferation (Alamar blue dye test), tenocyte synthetic activity
Frizziero et al., 2015	24 male Sprague-Dawley rats, aged 8 weeks, 280 ± 40 g body weight, Patellar tendon 18 rats run on a treadmill 1 h a day, three times a week for 10 weeks. The other six rats underwent no training	TRAINED GROUP (6 rats)–euthanized at the end of the 10-week training. Detrained group (12 rats)-euthanized after 10-week training + 4 weeks without exercise DETRAINED-HA GROUP (6 of the 12 detrained rats)-at the end of the 1, 2, 3, and 4 weeks without exercise a peri-patellar infiltration of 300 μL of 20 mg/2 mL HA (500–730 KDa, Fidia) was injected in the right patellar tendon DETRAINED-NaCl GROUP (remaining 6 of the 12 detrained rats)-at the end of the 1, 2, 3, and 4 weeks without exercise, at the level of the right patellar tendon, received a peri-patellar infiltration of 300 μL of physiological saline solution (Fresenius Kabi)	UNTRAINED GROUP (6 rats)-euthanized without training	TENDON ASSESSMENTS–modified semi-quantitative Movin grading scale (variables included: fiber structure, fiber arrangement, rounding of the nuclei, regional variation of cellularity, increased vascularity, collagen stainability, hyalinization), Tear density. ENTHESIS ASSESSMENTS-semi-quantitative score (variables included: patellar enthesis structure; cell morphology in calcified cartilage; cell morphology in non-calcified cartilage; chondrocyte cluster formation in calcified cartilage; chondrocyte cluster formation in non-calcified cartilage; tidemark integrity between calcified and non-calcified cartilage; matrix staining; vascularization).

**Table 2 jfmk-07-00003-t002:** Eight papers about clinical studies were selected from literature search.

Clinical Studies								
Article	Sample Description	Pathology	Injection Technique	Experimental Group	Control Group	Follow-Up	Outcome	Adverse Events
Meloni et al., 2007	Age = 31–71 N =56	Supraspinatus tendinosis (unresponsive to physical and medical therapy)	5 periarticularinjections (1/week) under ultrasoundguide	20 mg Sodium hyaluronate (MW = 500–730 KDa) together with 2 mL of 1% lidocaine and 2 mL of 0.9% sodium chloride solution	4 mL of 0.9% sodium chloride solution, together with 2 mL of 1% lidocaine	ultrasound exam controls at 3, 6 and 12 months from the last injection	Shoulder Range of motion (goniometer), Pain (VAS), degree of discomfort	There were no complications such as infections and no aggravations of symptoms compared with the pre-treatment state in either group
Blaine et al., 2008	Age = 50–79 N = 79	Shoulder pain due to glenohumeral joint osteoarthritis, rotator cuff tear (partial or complete), and/or adhesive capsulitis for at least 6 months but less than 5 years’ refractory to standard treatments (physical therapy, corticosteroid injections and the administration of oral pain medications)	5 intra-articular glenohumeralinjections (1/week)	5 INJECTION HYALURONATE GROUP-2 mL sodium hyaluronate (molecular weight, 500–730 kDa) 3 INJECTION HYALURONATE GROUP-three injections of sodium hyaluronate (500–730 KDa) followed by two injections of phosphate-buffered saline solution	CONTROL GROUP-2 mL injections of phosphate-buffered saline solution	7, 9, 13, 17, and 26 weeks after the initiation of therapy	Pain (VAS), maintenance of visual analogic scale pain relief, night pain improvement, patient global assessments, shoulder range of motion, general health questionnaire (short-form health survey-12)	All treatments were well tolerated. The most frequently reported adverse event considered to be related to study treatment was injection-site pain.
Frizziero et al., 2017	Age = 18–85 N = 34	Painful non-calcific rotator cuff tendinopathies confirmed by Instrumental diagnosis (US or MRI)	3 sub-acromial space injections (1/week) under ultrasound guidance	2 mL Hyaluronic acid (MW = 500–730 kDa)	4 weeks (1/week) low energy extracorporeal shockwave therapy (MODULITH^®^); each session consisted of 1600 shots at a frequency of 4 Hz. The applied maximum energy (0.15 mJ/mm^2^) was adjusted on the basis of the patient’s tolerance, Mean session duration: 10 min.	Parameters were evaluated at baseline (V0), at the end of the treatment (V1) and after 3 months of follow-up (V2)	Disabilities of the arm (DASH score and Constant-Murley scales)	No serious adverse events were recorded
Fogli et al., 2017	Age = 33–59 N = 62	lateral elbow (26), Achilles (34) or patellar (11) tendinopathies	3 peritendinous injections (1/week) under ecographic guide	2 mL Hyaluronic acid (MW = 500–730 kDa)		7, 14, and 56 days after first treatment	Pain (VAS), ultrasound assessment (changes in tendon thickness and neovascularization)	No serious adverse events were recorded
Frizziero et al., 2019	Age = 18–65 N = 35	Achilles (26) and patellar (9) mid-portion tendinopathies for ≥6 weeks duration and confirmed by ultrasound evaluation	3 peritendinous injection (1/week) between paratenon and tendon under Ecographic guide	20 mg/2 mL Hyaluronic acid (HyaloTend^®^, Fidia, MW = 500–730 KDa)		14, 45 and 90 days after the procedure	VISA-A and VISA-P at 90 days of follow-up, pain (NRS-11), US parameters (tendon appearance and neovascularization), Patient Global Assessment (PGA), Clinical Observer Global Assessment (COGA), rescue medication consumption (paracetamol) and Health-related Quality of Life (EuroQoL EQ-5D-5L questionnaire)	The treatment was well tolerated with only one adverse events in Achilles tendinopathy group, probably related to the injection procedure
Kanchanathepsak et al., 2020	Age = 43–69 N = 66	Trigger digits	1 peritendinous injection under ecographic guide	1 mL hyaluronic acid (500–730 kD, 20 mg/2 mL)	1 mL of 10 mg/mL of triamcinolone acetate	1, 3, 6 months	Residual symptoms (Quinell grading), Pain (VAS), disabilities of the Arm (DASH score)	There was no major complication found in the study. Three patients complained about local discomfort at 1 week after injection without any sign of local inflammation.
Mohebbi et al., 2021	Age = 16–70 N = 56	Rotator cuff tendinopathy (based on history, physical examination, and magnetic resonance imaging) with persistent shoulder pain for more than 6 weeks and less than 36 months	Single periarticular injection under ultrasound guidance	20 mg (2 mL) LMW-HA 1% (500–730 kDa)	20 mg (2 mL) HMW-HA 1% (>2000 kDa, Synogel, NikanTebKimia Pharmaceutical).	baseline, 1, 4 weeks, and 3 months after the interventions	Pain (VAS), shoulder range of motion (goniometer), quality of life (WHOQOL-Bref)	No serious adverse events were recorded. Nine patients in the HMW-HA group and 3 participants in the LMW-HA group showed signs of inflammation at the site of injection. Overall, the LMW-HA group showed more tolerance to injection pain
Rezasoltani et al., 2021	Age = 20–65 N = 51	Supraspinatus tendinopathy (based on history, physical examination, and magnetic resonance imaging) with persistent shoulder pain for more than 6 weeks and less than 3 months	Single subacromial injection under ultrasound guidance	20 mg (2 mL) LMW-HA 1% (500–730 kDa).	Physiotherapy group: 36 sessions, 3/week.	Baseline, 1, 4, and 12 weeks postintervention.	Pain (VAS), shoulder range of motion (goniometer), disabilities of the Arm (DASH score), quality of life (WHOQOL-Bref)	No important adverse events occurred in the two groups

## Data Availability

The study does not report any data.

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
