# Peer review of "Low Molecular Weight Hyaluronic Acid (500–730 Kda) Injections in Tendinopathies—A Narrative Review"

_jfmk, 2021, doi:10.3390/jfmk7010003_

Round 1

Reviewer 1 Report

The authors present a good updated review on intra-articular injections of hyaluronic acid for the treatment of tendinopathies. 

I would just like to make some suggestions:

  1. Add the following reference in the paragraph beginning on line 37: Sánchez Romero, E.A.; Pollet, J.; Martín Pérez, S.; Alonso Pérez, J.L.; Muñoz Fernández, A.C.; Pedersini, P.; Barragán Carballar, C.; Villafañe, J.H. Lower Limb Tendinopathy Tissue Changes Assessed through Ultrasound: A Narrative Review. Medicina 202056, 378.
  2. Add the following reference in the paragraph beginning on line 42: Muñoz Fernández AC, Barragán Carballar C, Villafañe JH, Martín Pérez M, Alonso Pérez JL, Díaz-Meco R, García Jiménez D, Sánchez Romero EA. A new ultrasound-guided percutaneous electrolysis and exercise treatment in patellar tendinopathy: three case reports. Front Biosci (Landmark Ed). 2021 Nov 30;26(11):1166-1175. 
  3. Is it possible to increase the letter size and improve figure 1?
  4. In the table of the articles, an explanatory table footnote must be added.

Reviewer 2 Report

This is an interesting article about hyaluronic acid injection in tendinopathies. Topic is hot in literature and the article is well written. 

Major suggestions:

I guess you wanna do a systematic review, so you need to improve the structure of your manuscript

  1. Title: "systematic review" should be included in your title
  2. you need to follow PRISMA (Preferred Reporting Items for Systematic Reviews and Meta-Analyses) guidelines
  3. State: "Each database was searched according to its specific syntax rules. Moreover, manual search of published and unpublished studies (conference abstracts, textbooks, “grey” literature) was also conducted and reference lists of retrieved articles were screened."
  4. you need to use quality assessment of selected studies - "Randomized clinical trials have been assessed by the revised Cochrane risk-of-bias tool (RoB 2.0) /  Non-randomized studies of intervention have been analyzed by the risk of bias in non-randomized studies of interventions assessment tool (ROBINS-I)" --> do not forget to include also this table
  5. statistical analysis: have a look at statistical paragraph in https://www.mdpi.com/2411-5142/5/2/33/htm

Minor suggestions:

  • keywords: hyaluronic acid (delete low molecular weight) --> more general could be better
  • examples of adductors tendinopathy: (http://rua.ua.es/dspace/handle/10045/108922) 
    and Achilles tendon: (http://rua.ua.es/dspace/handle/10045/110845)  
  • about histopathology, please consider the following reference https://www.researchgate.net/profile/Stefano-Palermi/publication/344597109_Influence_of_Supplements_and_Drugs_used_for_the_Treatment_of_Musculoskeletal_Disorders_on_Adult_Human_Tendon-Derived_Stem_Cells/links/5f833649a6fdccfd7b595d8c/Influence-of-Supplements-and-Drugs-used-for-the-Treatment-of-Musculoskeletal-Disorders-on-Adult-Human-Tendon-Derived-Stem-Cells.pdf  
  • line 44: extracorporeal shock wave therapy (ESWT)
  • line 49: autologous (not capital letter)
  • pay visually attention to the layout and to paragraphs!!
  • line 66: since 1990 (it is better)
  • Where are limitations of you study?? Please add a parargraph

Reviewer 3 Report

The paper is well written and the methodological approach is well conducted. Some minor issues:

1) please use the proper PRISMA flow diagram for Figure 1  (it can be downloaded here: http://prisma-statement.org/prismastatement/flowdiagram.aspx) and cite the relevant paper. 

2) the PRISMA flow diagram for depicting study selection should be part of the method section (an "evidence systensis" paragraph) rather than being included in the results section. 

3) line 27-28: please add at least one eminent reference for each of the tendons you list here (for the Achilles tendon, for instance, you may want to refer to these guidelines: http://dx.doi.org/10.32098/mltj.03.2018.03) 

Round 2

Reviewer 2 Report

Thanks for your response and for your modifications.

Good luck for your paper!